# Effect of Copper Tailing Content on Corrosion Resistance of Steel Reinforcement in a Salt Lake Environment

**DOI:** 10.3390/ma12193069

**Published:** 2019-09-20

**Authors:** Liming Zhang, Jia Li, Hongxia Qiao

**Affiliations:** 1School of Civil and Architectural Engineering, Nanchang Institute of Technology, Nanchang 330099, China; 2Library, Nanchang Institute of Technology, Nanchang 330099, China; zlm505506533@126.com; 3School of Civil Engineering, Lanzhou University of Technology, Lanzhou 730050, China; 2015994571@nit.edu.cn

**Keywords:** copper tailing, Qinghai Salt Lake, constant-current acceleration, corrosion resistance

## Abstract

With the increasing proportions of copper tailings of concrete in the Qinghai Salt Lake area of China, there arises the problem of corrosion of steel reinforcement in concrete structures. In this study, we determine the corrosion rate (C_R_), crack width, and corrosion potential of the steel reinforcement with copper tailing. This was achieved by conducting the constant-current accelerated corrosion test with different proportions of copper tailing in the brine environment of the Qinghai province. The results demonstrate that the corrosion potential (E_corr_) and the passivation area of the polarization curve decrease with the increase in the corrosion time, and the corrosion rate and crack width increase with the increase in the corrosion time. When the corrosion time is the same, the corrosion potential, crack width, and corrosion depth of the reinforcement decrease first and then increase with the increase in the copper tailing powder content. When the copper tailing powder content is 20%, the above parameters reach the minimum value. In the salt lake environment of Qinghai, China, the copper tailing powder content is recommended to be 20%.

## 1. Introduction

Copper tailings are the residues that remain after the crushing, pickling, and multiple screening of copper ores. These residues have a particle size of less than 0.3 mm. According to relevant data, the Jiangxi province alone has an annual copper tailing heap stock of 496 million tons [1], and the main copper tailings are stored in a reservoir dam (see Figure 1) [2,3], which poses a great threat to the surrounding environment and the safety of residents. Every year, the copper mining enterprises of the Jiangxi province of China spend more than 0.2 billion dollars on the construction, expansion, reinforcement, and health monitoring of tailing dams, and pay an environmental protection tax of more than 1.45 billion dollars. Some non-ferrous metal enterprises have been forced to stop production because they are unable to build new tailing dams. Therefore, it is of utmost importance to recycle the waste copper tailings in the Jiangxi province of China. Recently, research has mainly focused on using copper tailings as fine aggregate [4,5,6], mineral admixture [7,8,9,10], and aerated concrete products [11,12,13]. Owing to problems, such as low substitution amount and low product quality, the application of the abandoned copper tailings is still in the research stage and has not been widely applied. The results of the chemical composition analysis of the Ruichang copper tailings revealed that the content of pozzolanic active substances, such as SiO_2_, Fe_2_O_3_, and Al_2_O_3_, exceeds 70%; therefore, these tailings can be used as mineral admixtures.

Reinforced concrete is widely used as the basic material in the construction industry. Owing to the effects of high concentrations of chloride salts, the service life of reinforced concrete in the Qinghai Salt Lake area of China is less than 20 years [14,15,16]. Therefore, improving the corrosion resistance of reinforced concrete in the Qinghai Salt Lake area has become a difficulty in domestic and foreign research [17,18,19].

Foreign scholars [20,21] add micro-active mineral admixtures into geopolymers to improve the rheological reaction of the geopolymers, so as to prepare materials suitable for 3D printing. Onuaguluchi, O [22] have focused on the preparation of concrete as a mineral admixture and studied its durability in a chloride salt environment. They found that when the content of copper tailing slag is 10% of the cementing material, the durability of concrete can be improved, but beyond this content, the durability is deteriorated. Thomas, B.S [23] show that the copper tailing concrete (copper tailings up to 60% substitution of fine aggregate) exhibited good strength and durability characteristics.

The corrosion resistance of the copper tailing reinforced concrete in the Qinghai Salt Lake area is mainly caused by the corrosion of the steel reinforcement. Most scholars have studied the corrosion of steel reinforcement using the electrified acceleration method, in which the acceleration process usually adopts an immersion method [24,25,26,27]. In view of the above problems, this study adopted the constant-current acceleration test to comprehensively analyze the effect of the copper tailings concrete on the corrosion of the steel reinforcement by means of the polarization curve, crack observation, chloride ion content test, and X-ray energy spectrum analysis.

## 2. Test

### 2.1. Raw Materials

The copper tailings were obtained from Jiangxi province, and their main chemical compositions were SiO_2_, Al_2_O_3_, and CaO, with a small amount of Fe_2_O_3_ and MgO, as summarized in Table 1. To ensure that the fineness of the copper tailing met the requirements of the specification, a special ball mill was used to grind the tailing for 45 min. After grinding, the specific surface area was 410 m^2^/kg, and the particle size distribution is illustrated in Figure 2. The cement used was 42.5 ordinary Portland cement, and its chemical composition is listed in Table 1. The gravel used was 5–16 mm continuously graded gravel, and the water used was tap water. The concrete test mix ratio and performance of each group are presented in Table 2. The steel plate used was HRB400, which had a diameter of 12 mm.

Figure 2 shows SEM (scanning electron microscope) micro-graphs and the XRD (X-ray diffraction) patterns of copper tailing. Figure 2a presents a micro-graph of copper tailing. The majority of grains have an ellipsoidal and flaky shape. The rest of the copper tailing have a spherical shape. The grain surface is smooth. As shown in Figure 2b, the main materials of copper tailings are quartz (SiO_2_), and radite (Ca_3_Fe_2_Si_3_O_12_), gypsum (CaSO_4_), which is in accordance with the chemical analysis result that most of the minerals included are rich in SiO_2_ and Fe_2_O_3_.

Figure 3 shows the particle size distribution of the raw materials. Figure 3a shows that the particle size range of the copper tailings is 0–48.1 μm and the mean particle size (d_m_) and Hummel modulus (m_H_) of it is 9.0 μm and 22.54 based on the research of Katzer, J. [28]; the particle size range of cement is 0–100 μm and the mean particle size (d_m_) and Hummel modulus (m_H_) of it is 27.3 μm and 22.54. The mean particle size (d_m_) of copper tailings is 0.33 times that of cement. The Hummel modulus (m_H_) of copper tailing is 0.67 times that of cement. The maximum particle size of copper tailing is 0.48 times a of cement. Therefore, copper tailing can be filled in the pores of ordinary Portland cement [29,30,31,32].

### 2.2. Test Scheme

Four concrete mixing ratios were designed in this experiment, denoted as P, P10, P20, and P30, respectively, as listed in Table 2.

Each mixing ratio was composed of six 100 mm × 100 mm × 100 mm reinforced concrete specimens. A steel bar of length 100 mm was buried in the center of the concrete, with one end exposed 25 mm from the upper surface of the concrete. The other end was 25 mm from the bottom of the concrete, and the protective layer thickness of the specimen was set to 25 mm.

The mold was removed after 24 h once the specimen had formed, and the specimen was placed in an environment having a temperature of 20 + 2 °C and relative humidity of 95% RH (Relative Humidity) for more than 28 d. Then, the sample was configured to simulate the brine of the Qinghai Salt Lake, and a PS-3002D galvanostat was used to conduct the constant-current accelerated corrosion test. When conducting the electrification acceleration test, the current density was set at 200 μA/cm^2^, and the corresponding constant-current was 20 mA.

Chemical composition of Qinghai Salt Lake in China is shown as Table 3.

### 2.3. Test Method

#### Electrochemical Testing Methods

An electrochemical workstation was used for the electrochemical testing, in which the reference electrode used was the saturated calomel electrode, and the auxiliary electrode foil tape and steel bar in the reinforced concrete specimen were used as the working electrodes. The polarization curve scanning range was −200−200 mA, scanning rate was 334 mV/s, and frequency was 0.33 Hz. The polarization curve was measured every 103 h.

According to Faraday’s theorem (1) [33], the constant-current acceleration time required for each 1% increase in the theoretical mass loss rate under the action of a current of 4 mA can be obtained.

(1)$$t=\frac{∆m\cdot Z\cdot F}{M_{Fe}\cdot I}.$$

Here, *t* is the electrification time (s); ∆*m* is the quality loss of reinforcement (g); *Z* is the chemical valence of the reaction electrode (+2); *F* is Faraday’s constant (96,500 c/mol); *M_Fe_* is the atomic weight of iron 56 (g/mol); *I* is the current intensity (20 × 10^−3^ A). The data collection was conducted in cycles of 103 h durations until cracks of thickness 0.2 mm or greater appeared on the surface of the concrete specimen, after which the test was stopped.

## 3. Results and Discussion

### 3.1. Polarization Curve Results and Analysis

Figure 4a–d illustrates the polarization curves of the copper tailing reinforced concrete specimens P (normal concrete), P10 (The quality of copper tailing is 10% of that of cementing materials), P20 (The quality of copper tailing is 20% of that of cementing materials), and P30 (The quality of copper tailing is 30% of that of cementing materials), respectively, after being subjected to the constant-current accelerated corrosion for 0, 103 h, 206 h, 309 h, and 412 h. As illustrated in Figure 4a–c, the corrosion potential exhibited a negative trend with the extension of the accelerated constant-current corrosion, and the polarization curve passivation area gradually narrowed. In Figure 4d, the corrosion potential exhibited a positive trend at 309 h and 412 h. Therefore, we can conclude that the corrosion potential presents a negative trend when the copper tailing powder content is less than 20% of the total cement mass.

At the same corrosion time, the corrosion potential decreases first and then increases with the increase in the copper tailing powder content. Figure 4a–d indicates that the addition of 20% copper tailing powder had a good inhibition effect on the corrosion of the steel reinforcement at the initial stage of the electrochemical corrosion. As illustrated in Figure 4a, the corrosion potential exhibits a negative trend with the extension of the constant-current accelerated corrosion time. Figure 4b illustrates that the corrosion potential of group P10 gradually moved to the negative direction and the corrosion current density moved to the positive direction with the continuous increase of the constant-current accelerating corrosion time, indicating that the reinforcement corrosion becomes more severe with the extension of the constant-current accelerating time. As illustrated in Figure 4c, the corrosion potential of group P20 demonstrated a negative trend with the increase of the corrosion time under constant-current acceleration, but the change in the polarization curve was negligible after 103 h under constant-current acceleration. As illustrated in Figure 4d, the corrosion potential exhibits a negative trend with the extension of the constant-current accelerated corrosion time, but exhibits a positive trend at 206, 309, and 412 h. It can be concluded that the accumulation of early corrosion products on the surface of reinforced concrete specimens mixed with 10% copper tailings may temporarily hinder the development of corrosion. The above results demonstrate that, in the electrochemical corrosion process, when the content of copper tailing powder is less than 20% of the total cement mass, the inhibition effect on the steel corrosion is better.

Figure 5a–c illustrates the SEM photograph of the copper tailing reinforced concrete specimens P10, P20, and P30, respectively, after being subjected to the constant-current for 206 h. The columnar crystal in the concrete hole increases with the increase of the copper tailing content. When the copper tailing content is 20%, the columnar crystal fills the concrete hole and enhances the compacted concrete. When the content of copper tailing is 30%, the columnar crystals destroy the cement-based material structure around the concrete hole. Figure 5a illustrates that concrete P10 generates chips like bar crystals in holes and grows outward in the direction of holes. Figure 5b illustrates that the crystals generated in concrete P20 are short and thick with irregular distribution, and some honeycomb corrosion products are distributed on the gel surface. As Shown in Figure 5c, Concrete P30 has a large number of needle-rod crystals growing out of the gel, which makes the gel appear layered. The needle-rod crystals gather together and have a high degree of directivity.

The polarization curve is the relation curve between the electrode potential (E) and measured current (I) in the coordinate system in the electrochemical nondestructive testing method and the measured current density [34]. The corrosion of the steel bar is measured by the polarization curve and the corresponding corrosion potential (E_corr_), corrosion current density (i_corr_), corrosion rate (C_R_), and electrochemical etching parameters (E_tc_). The corresponding relation between the corrosion current density and instantaneous corrosion degree of steel reinforcement is given in literature [35], and is presented in Table 4.

Table 5 presents the corrosion electrochemical parameters corresponding to Figure 3a–d. As presented in Table 5, the corrosion current density and corrosion degree of the reinforced concrete specimens with copper tailing powder increase with the increase in electrification time. At 412 h of constant-current acceleration, the corrosion current density of the reinforced concrete specimens in Group P increased from 0.007 to 1.078 μA·cm^−2^, which is an increase of 154 times.

The corrosion current density of the reinforced concrete specimens in Group P10 increased from the initial 0.008 to 0.724 μA·cm^−2^, which is an increase of 90.5 times. It can be seen from Table 4 that the specimens in this group were moderately corroded at this time. The corrosion current density of the reinforced concrete specimens in Group P20 increased from 0.018 to 0.397 μA·cm^−2^, which is an increase of 22 times. It can be seen from Table 4 that the specimens in this group were slightly corroded at this time. The corrosion current density of the reinforced concrete specimens in group P30 increased from 0.012 to 1.334 μA·cm^−2^, which is an increase of 111 times. It can be seen from Table 4 that the specimens in this group were highly corroded at this time. According to the above analysis, after 412 h of constant-current accelerated corrosion, the order of the degree of corrosion of the steel reinforcement in the specimens from high to low was P30 > P > P10 > P20. This test result is consistent with that presented by Zhang Yunsheng and Sun Wei et al. [36], who suggested that the double-incorporation and single-incorporation of mineral admixtures can significantly improve the corrosion resistance of the steel bars in concrete. Additionally, this test result was consistent with the view proposed by Gao Xiangbiao [37], who suggested that the single-incorporation of mineral powder and fly ash can improve the protection performance of the steel bars in concrete.

### 3.2. Macroscopic Morphology of Concrete

As illustrated in Figure 5a–d, the crack width of the reinforced concrete with copper tailing powder increased with the increase in the constant-current accelerated corrosion time. When the corrosion time was 412 h, the reinforced concrete with copper tailing powder reached the failure limit and the crack width first decreased and then increased with the increase in the copper tailing powder content. The crack width was the smallest when the copper tailing content was 20%.

Figure 6a depicts that the maximum crack widths of the specimens in Group P are 0.15 and 0.44 mm, respectively, at 309 and 412 h of constant-current accelerated corrosion. According to the provisions in code for the design of concrete structures, which state that the maximum crack width of reinforced concrete members in the second and third class environments is 0.20 mm, it is considered that this group of specimens reaches the failure limit when accelerating for 412 h.

Figure 6b depicts that the maximum crack widths of specimens in Group P10 are 0.15 and 0.22 mm, respectively, at 309 and 412 h of constant-current accelerated corrosion.

Figure 6c depicts that the maximum crack width of specimens in Group P20 are 0.07 and 0.20 mm, respectively, at 309 and 412 h of constant-current accelerated corrosion.

Figure 6d depicts that the maximum crack width of specimens in Group P30 is 0.14 and 0.2 6 mm, respectively, at 309 and 412 h of constant-current accelerated corrosion.

The above results indicate that the specimens in all the groups reach the failure limit at 412 h of accelerated corrosion. In comparison with the specimens in Group P, the cracks in Groups P10, P20, and P30 were narrower, which implies that the incorporation of copper powder reduced the occurrence and development of cracks. The maximum crack widths corresponding to the specimens in Groups P, P10, P20, and P30 at 412 h of constant-current accelerated corrosion were P (0.44 mm), P30 (0.26 mm), P20 (0.22 mm), and P10 (0.20 mm), from the largest to the smallest, which were consistent with the results of the electrochemical polarization curve test and analysis.

### 3.3. Reinforcement Micro-Structure

To further study the morphology of the corroded steel bars in copper tailing reinforced concrete, EDS (energy-dispersive X-ray spectroscopy) equipped with the FEG-450 thermal field emission scanning electron microscope was used to conduct the linear scanning analysis of the corroded steel bars in Groups P, P10, P20, and P30. The results are illustrated in Figure 7, Figure 8, Figure 9 and Figure 10.

As can be seen from Figure 7a, at a depth of 0–90 μm, the concentration of O (2#) was high and fluctuated unsteadily and the concentration of Fe (1#) was also high, indicating severe corrosion of the reinforcement. Corresponding to the lighter rectangular area in Figure 6b, the corrosion of the reinforcement surface was severe. At a depth of about 90–160 μm, the concentration of O decreased, indicating a reduction in oxidation, corresponding to the darker region in Figure 7b, indicating that this part of the steel was not sufficiently corroded. At a depth of about 160–220 μm, the concentration of O was close to that at the depth of 90–160 μm, which implies that the degree of corrosion of the steel reinforcement is close to that at the depth of 90–160 μm. Below a depth of 220 μm, the concentration of O decreased again and did not increase again, indicating that the corrosion depth of the corroded steel bar in the Group P specimen was 220 μm and the corrosion was particularly severe.

As can be seen from Figure 8a, the high concentrations of O (2#) and Fe (1#) in the depth range of 0–60 μm indicates severe corrosion of the reinforcement surface, corresponding to the darker region in Figure 7b. At a depth of 60–180 μm, the concentration of O decreased, indicating the reduction of oxidation, corresponding to the darker region in Figure 8b, indicating that this part of the reinforcement was not fully corroded. Below a depth of 180 μm, the concentration of O decreased again and did not increase again, indicating that the corrosion depth of the corroded steel bar in the Group P10 specimen was 180 μm and the corrosion was severe.

As can be seen from Figure 9a, the concentration of Fe (1#) increased and the concentration of O (2#) began to appear at the depth of 60 μm after the reinforcement in the accelerated specimen. The main corrosion products were Fe(OH)_3_, FeOOH, and Fe_2_O_3_. At a depth of about 120 μm, the Fe and O concentrations were reduced, indicating insufficient oxidation, corresponding to the darker region in Figure 9b, indicating that this part of the reinforcement was not sufficiently corroded. At a depth of about 160 μm, the concentration of O decreased and did not increase, indicating that the steel bars at a depth of 160 μm and above were uncorroded. This corresponds to the color boundary in Figure 9b (i.e., the boundary between the corroded and uncorroded steel bars). From the comprehensive analysis of Figure 8a,b, the corrosion depth of the corroded steel bar in Group P20 was 160 μm and the corrosion was relatively serious.

As can be seen from Figure 10a, the high concentrations of O (2#) and Fe (1#) at depths ranging from 30 to 190 μm indicate severe corrosion of the reinforcement. Corresponding to the darker regions in Figure 10b, the corrosion of the reinforcement surface was severe. Below the depth of 190 m, the concentration of O again decreased and did not increase again, indicating that the corrosion depth of the corroded steel bar in Group P30 was 190 μm and the corrosion was serious.

To sum up, the reinforcement corrosion depths corresponding to the specimens in Groups P, P10, P20, and P30 at 412 h of constant-current accelerated corrosion, from largest to smallest, were P (220 μm), P30 (190 μm), P10 (180 μm), and P20 (160 μm). When the corrosion time was 412 h, the reinforcement corrosion depth first decreased and then increased with the increase in the copper tailing powder content. The reinforcement corrosion depth was the least when the copper tailing content was 20%.

## 4. Conclusions

By conducting the constant-current accelerated corrosion test of reinforced concrete with different copper tailing in the brine environment of the Qinghai province, the corrosion rate, crack width, and corrosion potential of the steel reinforcement with copper tailings were studied, and the following conclusions were obtained:The corrosion potential exhibited a negative trend with the extension of the accelerated constant-current corrosion, and the polarization curve passivation area gradually narrowed. At the same corrosion time, the corrosion potential decreases first and then increases with the increase in the copper tailings powder content. When the content of copper tailing powder is less than 20% of the total cement mass, the inhibition effect on the steel corrosion is better.The crack width of the reinforced concrete with copper tailing powder increased with the increase in the constant-current accelerated corrosion time. The reinforced concrete with copper tailing powder reached the failure limit, and the crack width first decreased and then increased with the increase in the copper tailing powder content. When the copper tailing content is less than 30%, the crack width was the smallest with 20% copper tailing content of cement quality.The corrosion current density and corrosion degree of the reinforced concrete specimens with copper tailing powder increase with the increase in electrification time. After 412 h of constant-current accelerated corrosion, the degree of corrosion of the steel reinforcement in the specimens first decrease then increase with the increase of copper tailing powder content. The degree of corrosion of the steel reinforcement was the lowest when the copper tailing content was 20%.In the reinforced concrete structure of the Qinghai Salt Lake environment, the recommended content of copper tailings is 20%.

## Figures and Tables

**Figure 1 materials-12-03069-f001:**
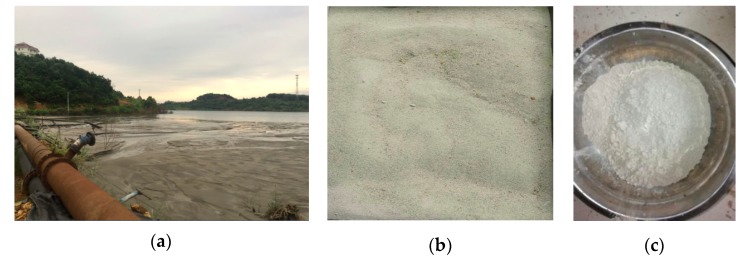
Copper tailing powder in copper tailing reservoir. (**a**) Reservoir of copper tailing; (**b**) copper tailing; (**c**) copper tailing power.

**Figure 2 materials-12-03069-f002:**
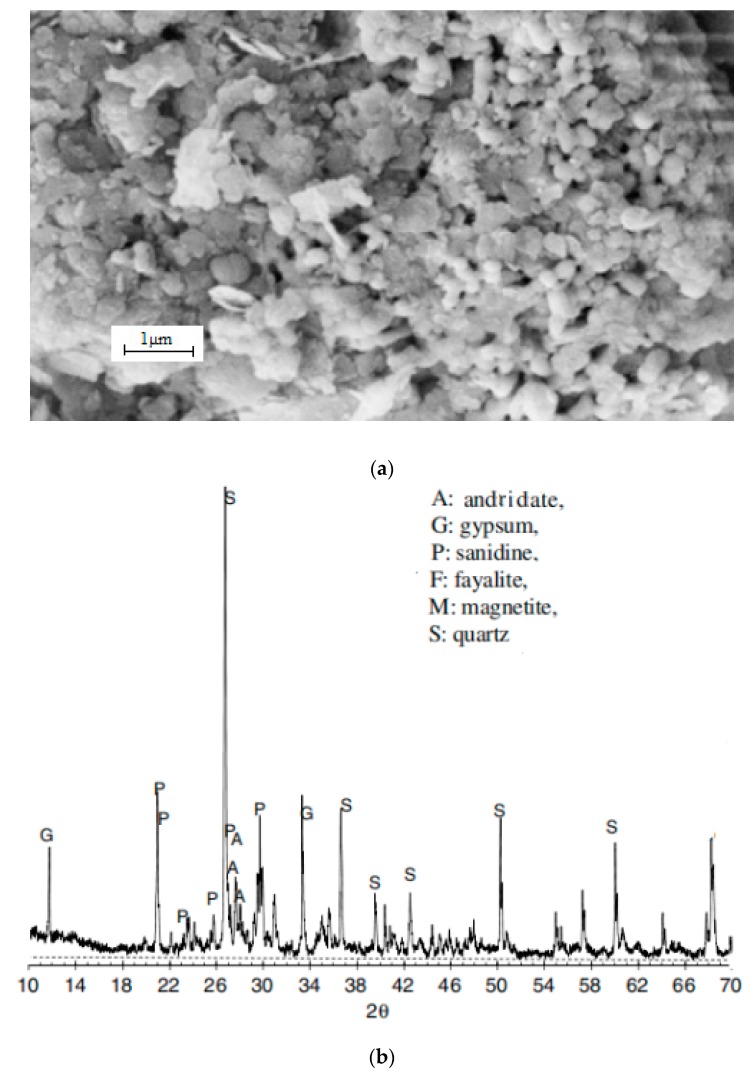
SEM (scanning electron microscope) and the XRD (X-ray diffraction) of copper tailings: (**a**) SEM micro-graphs of copper tailings; (**b**) XRD pattern of copper tailings.

**Figure 3 materials-12-03069-f003:**
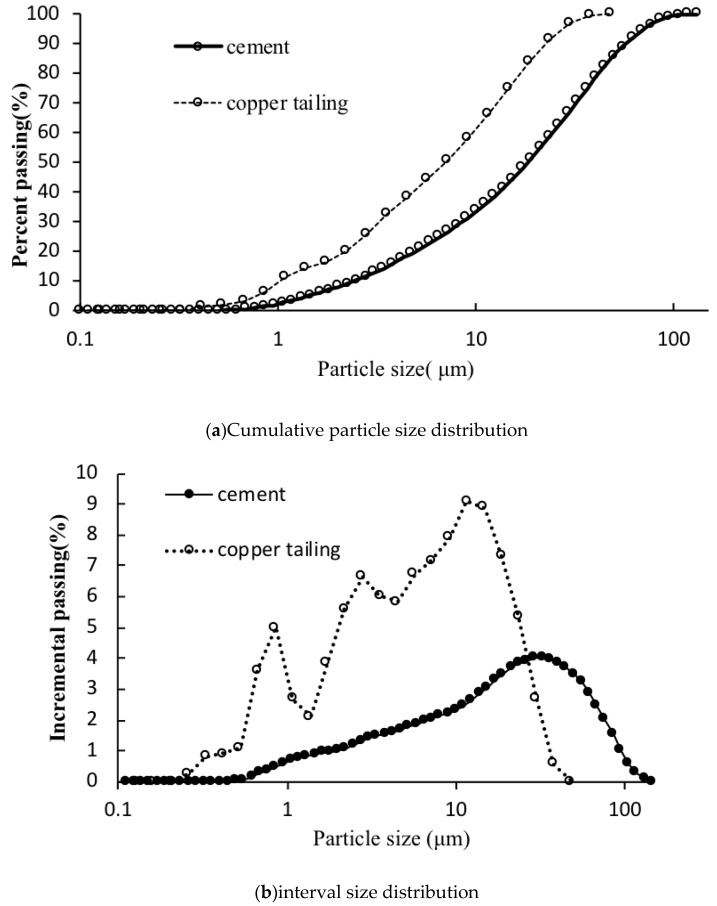
Particle size distribution of raw materials: (**a**) Cumulative particle size distribution; (**b**) interval size distribution.

**Figure 4 materials-12-03069-f004:**
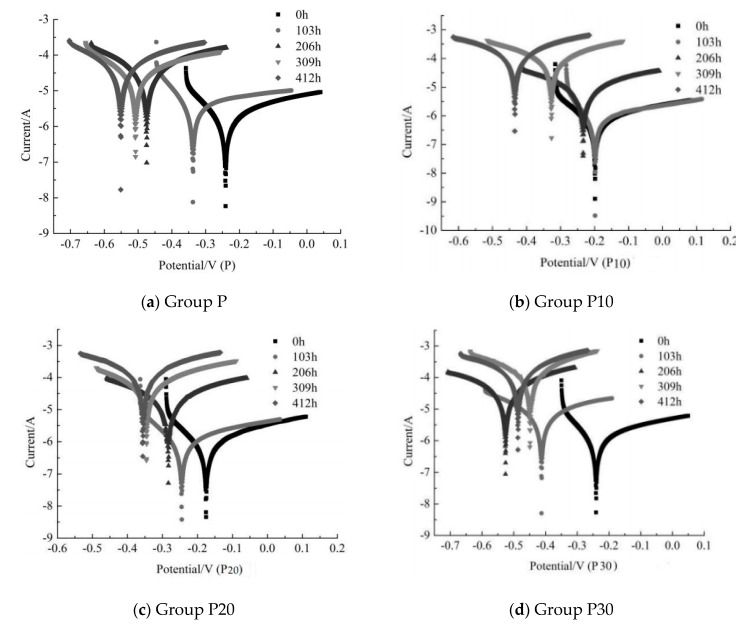
Polarization curves of reinforced concrete with different copper tailings. (**a**) Polarization curves of Group P; (**b**) polarization curves of Group P10; (**c**) polarization curves of Group P20; (**d**) polarization curves of Group P30.

**Figure 5 materials-12-03069-f005:**
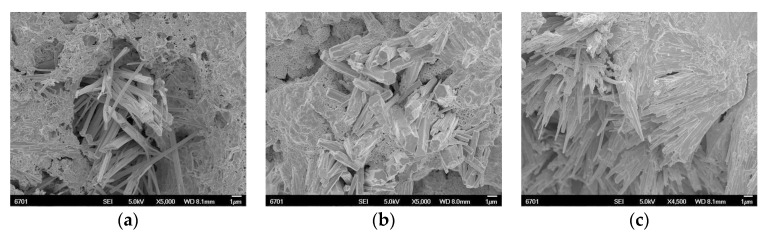
SEM photograph of reinforced concrete with different copper tailings with 206 h corrosion time. (**a**) Group P10; (**b**) Group P20; (**c**) Group P30.

**Figure 6 materials-12-03069-f006:**
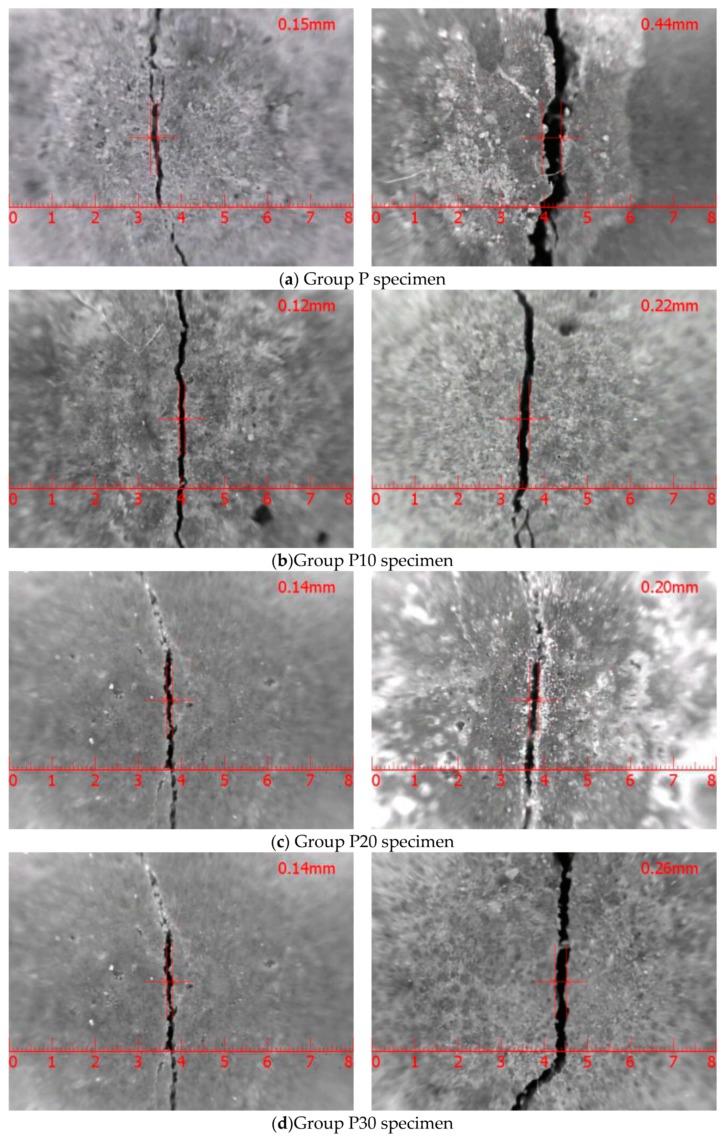
Effect of copper tailing powder content on crack width of concrete. (**a**) Constant-current acceleration test piece of Group P specimen; (**b**) constant-current acceleration test piece of Group P10 specimen; (**c**) constant-current acceleration test piece of Group P20 specimen; (**d**) constant-current acceleration test piece of Group P30 specimen.

**Figure 7 materials-12-03069-f007:**
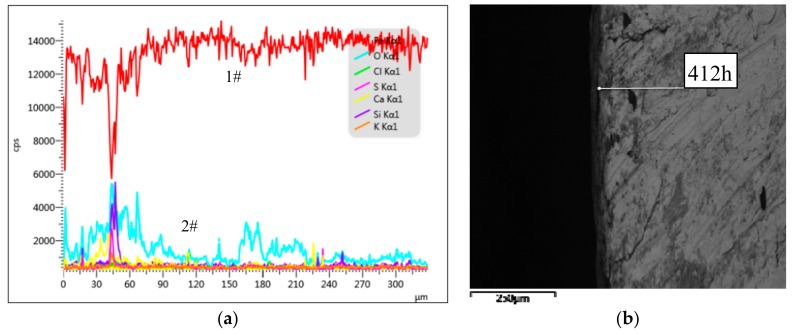
Group P line scan results. (**a**) The structure of steel corrosion products; (**b**) Micro-structure of corroded steel bars.

**Figure 8 materials-12-03069-f008:**
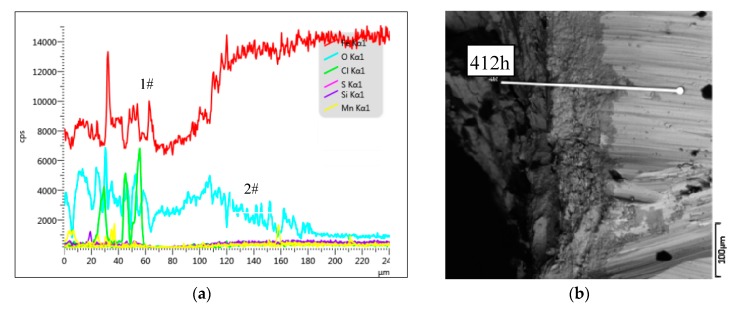
Group P10 line scan results. (**a**) The structure of steel corrosion products; (**b**) Micro-structure of corroded steel bars.

**Figure 9 materials-12-03069-f009:**
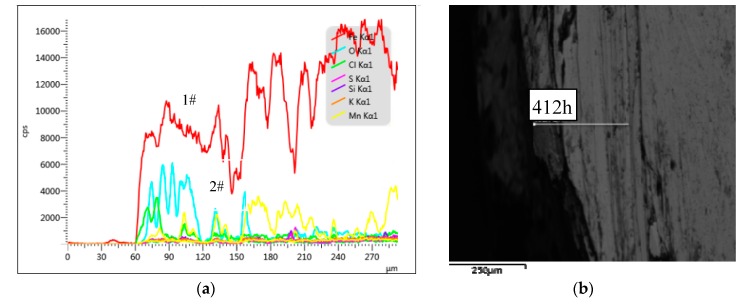
Group P20 line scan results. (**a**) The structure of steel corrosion products; (**b**) Micro-structure of corroded steel bars.

**Figure 10 materials-12-03069-f010:**
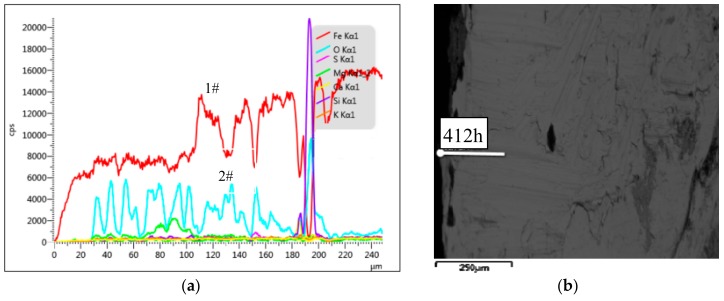
Group P30 line scan results. (**a**) The structure of steel corrosion products; (**b**) Micro-structure of corroded steel bars.

**Table 1 materials-12-03069-t001:** Cementitious material chemical composition.

Binder Material Type	SiO_2_	Al_2_O_3_	Fe_2_O_3_	CaO	MgO	SO_3_	Loss
Cement	66.5	5.5	3.3	15.7	1.7	2.0	5.3
Copper tailing	58.5	6.6	15.8	12.7	2.8	3.2	0.4

**Table 2 materials-12-03069-t002:** Concrete mixing ratios (kg/m^3^)

No.	Raw Material Quality (kg/m^3^)					Slump/mm	28 Day Compressive Strength/MPa
	Cement	Water	Sand	Coarse Aggregate	Copper Tailing		
P	450	158	634	1167	0	98	46.6
P10	405	158	634	1167	45	100	50.3
P20	360	158	634	1167	90	105	50.6
P30	315	158	634	1167	135	118	45.1

**Table 3 materials-12-03069-t003:** Chemical composition of Qinghai Salt Lake, China.

Ion Name	Na^+^	Mg^2+^	K^+^	Ca^2+^	Cl^−^	SO_4_^2−^	CO_3_^2−^	HCO_3_^−^
Unit (mg/dm^3^)	68.36	35.13	5.98	4.24	204.21	22.29	0.17	0.13

**Table 4 materials-12-03069-t004:** Relationship between corrosion current density (i_corr_) and corrosion degree of steel.

*i*_corr_/(μA·cm^−2^)	*i*_corr_ < 0.2	0.2 < *i*_corr_ < 0.5	0.5 < *i*_corr_ < 1.0	1.0 < *i*_corr_ <10	*i*_corr_ > 10
Corrosion degree	Passivation state	Low corrosion condition	Moderate corrosion condition	High corrosion condition	Extreme corrosion condition

**Table 5 materials-12-03069-t005:** Electrochemical parameters of the polarization curve.

No.	t (h)	0	103	206	309	412
P	E_corr_/V	−0.241	−0.337	−0.474	−0.508	−0.551
	i_corr_/(μA·cm^−2^)	0.007	0.009	0.066	0.726	1.078
	C_R_/(10^−3^ mm·a^−1^)	0.141	0.616	3.751	11	15.47
P10	E_corr_/V	−0.199	−0.198	−0.234	−0.327	−0.434
	i_corr_/(μA·cm^−2^)	0.008	0.011	0.172	0.446	0.724
	C_R_/(10^−3^ mm·a^−1^)	0.098	0.123	1.995	5.17	8.393
P20	E_corr_/V	−0.175	−0.245	−0.283	−0.346	−0.356
	i_corr_/(μA·cm^−2^)	0.018	0.040	0.299	0.256	0.397
	C_R_/(10^−3^ mm·a^−1^)	0.209	0.466	3.466	2.972	4.6
P30	E_corr_/V	−0.241	−0.412	−0.526	−0.449	−0.486
	i_corr_/(μA·cm^−2^)	0.012	0.053	0.323	0.948	1.334
	C_R_/(10^−3^ mm·a^−1^)	0.081	0.109	0.771	8.422	12.5

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
