# Peer review of "Effect of Copper Tailing Content on Corrosion Resistance of Steel Reinforcement in a Salt Lake Environment"

_materials, 2019, doi:10.3390/ma12193069_

Round 1
Reviewer 1 Report
The paper is within the scope of the journal. It deals with an interesting and current topic of using copper waste material to improve resistance of concrete against corrosion. The paper is worth publishing after addressing some minor issues:
Authors shouldn’t give the full commercial names of the used equipment or materials. A particular apparatus/material should be described in brief (giving its key characteristics) and, if needed, referred to literature with a thorough description of such an apparatus/material (e.g. line 90, line 97, line 99 etc.). Granulometric properties of all used materials (especially of copper tailing) should be given in detail. While discussing this topic in the paper Authors may find the following publication useful: Katzer, J. (2012) ‘Median diameter as a grading characteristic for fine aggregate cement composite designing’, Construction and Building Materials, 35. doi: 10.1016/j.conbuildmat.2012.04.050. Table 2: What does “stone” mean? Do Authors mean “coarse aggregate”? Table 2: Instead of “Mpa”, there should be “MPa”. Table 3: Instead of “mg/L”, there should be “mg/dm3”.Author Response
Comments and Suggestions for Authors The paper is within the scope of the journal. It deals with an interesting and current topic of using copper waste material to improve resistance of concrete against corrosion. The paper is worth publishing after addressing some minor issues: 1、Authors shouldn’t give the full commercial names of the used equipment or materials. Response: i receive your advice , Line 58:“The copper tailing were obtained from Ruichang City, Jiangxi province” delete Ruichang City Response : delete Ruichang City Line 62:“The cement used was Jiangxi Conch 42.5 ordinary Portland cement” delete Jiangxi Conch Response : delete Jiangxi Conch Line 63:“The gravel used was 5–16 mm continuously graded gravel from Anyi county,nanchang ”deletefrom Anyi county,nanchang Response : deletefrom Anyi county,nanchang 2、 A particular apparatus/material should be described in brief (giving its key characteristics) and, if needed, referred to literature with a thorough description of such an apparatus/material (e.g. line 90, line 97, line 99 etc.). Response :thank for your advice Line 90 “a PS-3002D galvanost was used to conduct the constant-current accelerated corrosion test with a current of 4 mA.” Response : delete PS-3002D Line 97“An electrochemical workstation from Zahner E, Germany was used for the electrochemical testing” Response : delete from Zahner E, Germany Line99 “electrode foil tape was 26PH made in SchottScienceLine,Germany and steel bar in the reinforced” Response : delete made in SchottScienceLine,Germany 3、Granulometric properties of all used materials (especially of copper tailing) should be given in detail. While discussing this topic in the paper Authors may find the following publication useful: Katzer, J. (2012) ‘Median diameter as a grading characteristic for fine aggregate cement composite designing’, Construction and Building Materials, 35. doi: 10.1016/j.conbuildmat.2012.04.050. Response:Figure3(a) shows that particle size range of copper tailing is 0-48.1μm and the mean particle size(dm) and hummel modulus(mH) of it has 9.0μm and 22.54 based on the research of Katzer, J. [24] ; the particle size range of cement is 0-100μm and the mean particle size(dm) and hummel modulus(mH) of it has 27.3μm and 22.54 .The mean particle size (dm) of copper tailing is 0.33 times of cement. The hummel modulus (mH)of copper tailing is 0.67times of cement. The maximum particle size of copper tailing is 0.48 times a of cement.Therefore, copper tailing can be filled in the pores of ordinary Portland cement [25-28]. 4、 Table 2: What does “stone” mean? Do Authors mean “coarse aggregate”? Table 2: Instead of “Mpa”, there should be “MPa”. Table 3: Instead of “mg/L”, there should be “mg/dm3”. Table 2:What does “stone” mean? Do Authors mean “coarse aggregate”? Response : yes , I use stone for coarse aggregate.i my article , i have already modified the stone into coarse aggregate of table2. Table 2: Instead of “Mpa”, there should be “MPa”. Response : yes, i have already used MPa instead of Mpa inTable2 . Table 3: Instead of “mg/L”, there should be “mg/dm3”. Response : yes .I will follow your advice and modify table2:mg/L tomg/dm3. Date of this review 27 Aug 2019 15:26:58

Reviewer 2 Report
The used copper tailing need to be characterized in terms of SEM and XRD.
Table 2. How the Test scheme is decided in this paper
Line 101: Ref missing for the setting used
Section 3.1 only the obatined results are described. The output is not linked with applied scintific causes. .
Fig. 4 some imnages can be improved for better visulaization.
On which section Reinforcement micro-structure was seen? Update it in the methodloogy.
Rewrite the conclusion with scientfic casues.
Introduction section can emphasize on sustainabilty aspects of copper tailing and other by products including some recent papers:
Reusing copper tailings in concrete: corrosion performance and socioeconomic implications for the Lefke-Xeros area of Cyprus
Extrusion and rheology characterization of geopolymer nanocomposites used in 3D printing
Strength and durability characteristics of copper tailing concrete
Rheological behavior of high volume fly ash mixtures containing micro silica for digital construction application
Author Response
The used copper tailing need to be characterized in terms of SEM and XRD.
Response :Fig. 2 shows SEM micro-graphs and the XRD patterns of copper tailing. Fig. 2(a) presents a micro-graph of copper tailing . The majority of grains have an ellipsoidal and flaky shape. The rest of copper tailing have a spherical shape. the grain surface is smooth.As shown in Figure 2(b),The main materials of copper tailing are quartz (SiO2), andradite(Ca3Fe2Si3O12), gypsum (CaSO4).in accordance with the chemical analysis result that most of the minerals included are rich in SiO2 and Fe2O3.
SEM micro-graphs of copper tailing
(b)XRD pattern of copper tailing
Figure 2. SEM and XRD of copper tailing (a) SEM micro-graphs of copper tailing (b)XRD pattern of copper tailing
2、Table 2. How the Test scheme is decided in this paper
Response :When conducting the electrification acceleration test, the current density is set at 200 microns A/cm2, and the corresponding constant current is 20mA.
3、Line 101: Ref missing for the setting used
Response :The polarization curves are measured every 103 hours.
4、Section 3.1 only the obatined results are described. The output is not linked with applied scintific causes. .
Response: Figures 4(a)–(c) illustrate the SEM photograph of the copper tailing reinforced concrete specimens P10, P20, andP30, respectively.after being subjected to the constant-current for206 h. The columnar crystal in the concrete hole increases with the increase of the copper tailings content. When the copper tailings content is 20%, the columnar crystal fills the concrete hole and enhances the compacted concrete.When the content of copper tailings is 30%, the columnar crystals destroy the cement-based material structure around the concrete hole.Figure 4(a) illustrates that concrete P10 generates chips like bar crystals in holes and grows outward in the direction of holes. Figure 4(b) illustrates that the crystals generated in concrete P20 are short and thick with irregular distribution, and some honeycomb corrosion products are distributed on the gel surface.As Shown in figure 4(c), Concrete P30 has a large number of needle-rod crystals growing out of the gel, which makes the gel appear layered. The needle-rod crystals gather together and have a high degree of directivity.
Group P10(b) Group P20 (c) Group P30
Figure 4. SEM photograph of reinforced concrete with different copper tailing with 206h corrosion time
5、Fig. 4 some imnages can be improved for better visulaization.
Yes .i have already do it
6、On which section Reinforcement micro-structure was seen?
Response : it is my mistake.Section 3.3“Reinforcement micro-structure” is revised “corroded steel bar micro-structure”.
7、Introduction section can emphasize on sustainabilty aspects of copper tailing and other by products including some recent papers:Reusing copper tailings in concrete: corrosion performance and socioeconomic implications for the Lefke-Xeros area of Cyprus
;Extrusion and rheology characterization of geopolymer nanocomposites used in 3D printing
Strength and durability characteristics of copper tailing concrete
Rheological behavior of high volume fly ash mixtures containing micro silica for digital construction application.
Response : I have already add some sustainabilty aspects of copper tailing paper in this article.
Foreign scholars[20-21] add micro-active mineral admixtures into geopolymers to improve the rheological reaction of the geopolymers, so as to prepare materials suitable for 3D printing.Onuaguluchi, OandEren, O[22–23] have focused on the preparation of concrete as a mineral admixture and studied its durability in chloride salt environment. They found that when the content of copper tailing slag is 10% of the cementing material, the durability of concrete can be improved, but beyond this content, the durability is deteriorated. Blessen SkariahThomas [24] the copper tailing concrete (Copper tailing up to 60% substitution of fine aggregate) exhibited good strength and durability characteristics.
Reference
Biranchi,Panda;CiseUn,luer;Ming,JenTana.Extrusion and rheology characterization of geopolymer nanocomposites used in 3D printing. Composites Part B: Engineering,2019,176, 1072-90. Biranchi,Panda;Ming,JenTan. Rheological behavior of high volume fly ash mixtures containing micro silica for digital construction application.Materials Letters,2019,237,348-351. Onuaguluchi, O; Eren, O .Reusing copper tailings in concrete: corrosion performance and socioeconomic implications for the Lefke-Xeros area of Cyprus.Journal of Cleaner Production,2016,112, 420-429. Onuaguluchi, O.; Eren, O. Strength and durability properties of mortars containing copper tailings as a cement replacement material. J. Environ. Civ. Eng,2013, 17, 19–31. Blessen Skariah,Thomas;AlokDamare;C.Gupta.Strength and durability characteristics of copper tailing concrete.Construction and Building Materials,2013,48, 894-900.

Round 2
Reviewer 2 Report
It can be accepted now for publication